# Effects of Academic Motivation on Clinical Practice-Related Post-Traumatic Growth among Nursing Students in South Korea: Mediating Effect of Resilience

**DOI:** 10.3390/ijerph17134901

**Published:** 2020-07-07

**Authors:** Mi Ra Yun, Eun Ju Lim, Boas Yu, Sookja Choi

**Affiliations:** 1Red Cross College of Nursing, Chung-Ang University, Seoul 06974, Korea; mirayun21@cau.ac.kr (M.R.Y.); dew7593@cau.ac.kr (E.J.L.); 2Henry P. Becton School of Nursing and Allied Health, Fairleigh Dickinson University, Teaneck, NJ 07666, USA; byu@fdu.edu

**Keywords:** motivation, posttraumatic growth, psychological resilience, nursing students

## Abstract

Post-traumatic growth (PTG) refers to personal growth that occurs after experiencing challenges. For many nursing students, PTG could occur during their clinical practice. Academic motivation and resilience could help students to overcome these traumatic clinical experiences and possibly achieve PTG. This study examined the relationships between nursing students’ academic motivation and resilience leading to post-traumatic growth. A total of 291 nursing students from three South Korean nursing colleges participated in this cross-sectional study. Self-report questionnaire data were analyzed using *t*-tests, ANOVA, correlations, and hierarchical multiple linear regressions. Intrinsically motivated students’ PTG scores were significantly higher compared to extrinsically motivated students (*t* = 4.62, *p* < 0.001). Resilience scores showed similar results (*t* = 3.81, *p* < 0.001). Significant total, direct, and indirect effects of academic motivation on resilience and PTG were found. In addition, resilience mediated 40.9% of the relationship between academic motivation and PTG. Nursing students with intrinsic academic motivation were more likely to achieve high PTG scores, and resilience mediated the relationship between academic motivation and PTG. It is suggested that the importance of academic motivation must be considered in the early stages of university-level nursing education to increase retention rates of nursing students.

## 1. Introduction

Nursing shortage poses a serious challenge to many countries worldwide, which impacts the quality of nursing care and patient outcomes, as well as the recruitment, training, and retention costs of nursing personnel [1]. In South Korea, the average attrition rate for nurses is estimated at 19.7%; and for new nurses with less than one year of employment, the rate is much higher and steadily increasing from 30.3% in 2011 to 33.9% in 2015, and 42.7% in 2017 [2]. In an attempt to address the nursing shortage, the South Korean government instituted a policy to increase nursing school admissions and to improve nurses’ work environment; however, the turnover rates are still increasing [3,4]. Up to this point, many research studies from South Korea and other countries tended to focus on factors related to nurse turnover and attrition rates [5]. However, it is imperative to examine the unique characteristics of nursing students to see what motivated them to enter nursing. There is a gap in knowledge when exploring the attributes of South Korean nursing students, and this exploration may lead to possible approaches to curtail high drop-out rates of nursing students and decrease high attrition rates of new nurses.

### 1.1. Background 

Nursing education aims to ensure students successfully undergo sufficient training and become qualified nurses. Unfortunately, nursing students often feel considerably stressed, due to the rigorous and intensive nature of the academic curriculum. Consequently, many are unsuccessful and drop out of the program. [6,7,8]. 

Although most nursing students experience similar stressful situations in their clinical practice environments, many differ considerably in their perceptions and reactions to these situations. Some students may feel traumatized by a stressful clinical experience, such as emergent crises or interpersonal conflicts; whereas others become more introspective and motivated from their encounter.

The term post-traumatic growth (PTG) explains these individual differences in personal growth among nursing students following traumatic events. Tedeschi and Calhoun [9] clarified that when confronting a challenge, individuals can achieve internal personal growth by overcoming it. Furthermore, PTG refers to a positive change in one’s self, interpersonal relationships, and life philosophy as a result of internal struggle with adversity [9]. 

The term vicarious post-traumatic growth (VPTG) refers to the indirect PTG that occurs when a person is exposed to a traumatic event through another source, rather than being exposed to it directly [10]. Currently, VPTG has been reported among a range of healthcare professionals [11] and could also be applicable to nursing students experiencing an adverse clinical practice. However, very limited literature on nursing students’ PTG or VPTG is currently available.

Another important concept in nursing education is academic motivation, which can be divided into intrinsic and extrinsic motivation [12,13]. Intrinsic motivation refers to performing an activity for personal pleasure and the satisfaction derived from the activity itself; it is characterized by feelings of excitement, interest, happiness, curiosity, and self-determination. By contrast, extrinsic motivation occurs in a response to external stimulation, such as the expectation of rewards or punishments, possibility of promotion in the workplace, or acquisition of a diploma [14,15]. Research reveals that when selecting their major, students with intrinsic motivation tend to have much higher satisfaction with their chosen majors, and demonstrate better career decision-making self-efficacy and employment preparation behavior than those with extrinsic motivation [12,16]. Since motivation is an important psychological concept that determines the level and intensity of behavior in education [13,17], it is important to examine the reasons why students choose nursing as a major and understand how PTG is related to nursing students’ clinical practice. 

Resilience in nursing students had been extensively studied in nursing education. Resilience is an individual characteristic and a positive force that prevents individuals from being overwhelmed by stress and conflict, enabling them to successfully overcome and adapt to adversity [18]. Research shows that nursing students’ resilience strengthens their PTG and ability to handle stress in clinical settings [19,20]. 

It is important to explore factors that may influence PTG among students who will soon become nurses, as it may contribute to the development of strategies to relieve high stress levels, improve their mental health and well-being, as well as produce and retain highly qualified nurses. Nursing educators also need to understand students’ initial motivation and reasons for selecting nursing as their major in order to better guide them through their education process [21]. In addition, educators should explore factors influencing nursing students’ positive personal growth and develop strategies to facilitate this growth. In the past, limited research focused on students’ motivation in selecting nursing as their major, and there is a very little research that has explored how students adapt and learn from clinical practice experiences based on their motivation and resilience, and how these affect their personal growth.

The purpose of this study was to investigate the relationships among PTG, academic motivation, and resilience in nursing students. Furthermore, we aimed to identify what factors led to nursing students’ PTG in the face of challenging situations during their clinical practice.

### 1.2. Research Objectives

This descriptive, cross-sectional study examined factors related to selection of nursing as a college major and explored the relationships between nursing students’ academic motivation, PTG, and resilience. Specifically, the study addressed the following research questions:Are there differences in PTG according to academic motivation?Is there a correlation between academic motivation, PTG, and resilience?Does academic motivation influence PTG and resilience?Does resilience mediate the relationship between academic motivation and PTG?

## 2. Methods

### 2.1. Participants

Third- and fourth-year nursing students who had experienced at least one year of clinical nursing practice in three nursing colleges of South Korea participated in this study. Those who had experienced less than one year of clinical practice were excluded. A total of 320 students were surveyed, following statistical analysis using the G * Power 3.1.3 program (Heinrich Heine University Düsseldorf, Düsseldorf, Germany) at a significance level (⍺) of 0.05, power (1 − β) of 0.90, and medium effect size of regression at 0.05. 

### 2.2. Data Collection

A convenience sample of 320 nursing students was recruited from three nursing colleges located in the South Korean cities of Seoul and Jeolla province. Questionnaires were distributed only to those students who had provided written informed consent. Of the 320 questionnaires that were distributed, 309 were returned. After excluding incomplete responses, we used 291 questionnaires in the final data analysis.

### 2.3. Measures

#### 2.3.1. Academic Motivation

Academic motivation is one of the most important psychological concepts in determining the level of drive and intensity of student behavior, and can be divided into intrinsic and extrinsic motivation [12,13]. 

To identify respondents’ reasons for selecting nursing as their college major and classify their academic motivation, we asked the following question: “What motivated you to choose nursing as your college major?” We considered intrinsic motivation and included items such as an aptitude and interest, a desire to serve and help other people, and a willingness to work in foreign countries or to learn/study continuously in the nursing field. Furthermore, we considered extrinsic motivation and included items such as ease of finding employment after graduation, preference or selection of college major based on high school grade point average, and recommendations from parents or relatives as a basis when choosing a college major.

#### 2.3.2. Post-Traumatic Growth (PTG)

PTG is defined as positive psychological growth and subjective perception changes resulting from an individual’s experience in highly challenging environments. In this study, we used the tools developed by Tedeschi and Calhoun [9], translated into Korean by Song, Lee, Park, and Kim [22]. The tool comprised five sub-domains: relating to others, identifying new possibilities, personal strength, spiritual change, and appreciation of life. Participants were asked to rate the extent to which they had experienced their growth, in each of 21 items on the 6-point Likert scale, ranging from 0 (did not experience) to 5 (experienced to a very high degree). Total scores can range from 0 to 105, with scores below 60 (mean score 2.85) indicating a low level of growth, scores between 60 and 79 (mean scores 2.85 and 3.76) indicating a moderate level of growth, and scores 80 (means score 3.77) or above indicating a high level of growth [23]. In this study we used mean scores. A higher means score indicates higher growth levels. In the current sample, Cronbach’s alpha was 0.94. The PTG has been translated into many languages including Korean. It has been widely used in research studies and tested in various populations with psychometric properties [9].

#### 2.3.3. Resilience

Resilience is a personal characteristic of positive internal force which prevents one from being overwhelmed by stress and adversity. It enables one to successfully overcome and adapt to such experiences [18]. In this study, we used the resilience tool developed by Shin et al. [18], comprising a total of 27 items in the following three sub-domains: self-control ability, interpersonal ability, and affirmation. Each items was scored on a 5-point Likert scale ranging from ‘1′ for ‘not at all’ to ‘5′ for ‘strongly agree’, with possible scores for resilience range from 27 to 135. Each participant was assigned a score equal to the mean of his or her scores on all items, with higher scores indicating better resilience. The Korean version of the resilience scale has been developed and psychometrically tested [18]. In this study, Cronbach’s alpha was 0.88.

#### 2.3.4. General Characteristics

Among respondents’ general characteristics, we examined age, gender, academic year, economic status, grade point average, clinical practice stress level, satisfaction with one’s major, and satisfaction with university life. All characteristics except age and gender were categorized into three groups. In a regression analysis, all categorical variables were divided into two groups (Table 1). 

#### 2.3.5. Ethical Considerations

This study obtained approval from the Institutional Review Board of Chung-Ang University (IRB No: 1041078-201601-HRSB-004-01). Participants were enrolled only after they provided written, informed consent. In addition, participants were informed that they could choose not to participate in the study at any time, without any personal disadvantage or penalty. All self-questionnaires were completed anonymously, and data on each participant’s information were coded. The collected data were maintained in the researchers’ private office, which was kept locked to ensure confidentiality.

### 2.4. Statistical Analysis

The collected data were analyzed using IBM SPSS Statistics 25.0 for Windows (IBM Corp., Armonk, NY, USA). We investigated the differences in resilience and PTG scores for various demographic characteristics using independent *t*-tests, analysis of variance, and Scheffè post hoc tests. Correlations among variables were analyzed using Pearson’s and Spearman’s correlation coefficients. To test the mediating effect of resilience, several hierarchical multiple linear regressions were performed, based on the procedures developed by Baron and Kenny [24] and method described by Hayes and Preacher [25]. The indirect effect of academic motivation on PTG facilitated by resilience was determined using bootstrapping. Furthermore, to test the mediating effects of resilience on the relationship between academic motivation and PTG, we used the following three-step mediation analysis method as described by Baron and Kenny: 

Step 1. Academic motivation predicts PTG (path *c*).

Step 2. Academic motivation predicts resilience (path *a*).

Step 3. Both academic motivation and resilience predict PTG (paths *b* and *c*).

The bootstrapping was used to estimate direct and indirect effects through a repeated sampling of data (*n* = 5000). Based on this analytical approach, the mediating effects of resilience on the relationship between academic motivation and PTG were suggested to have significant total and indirect effects. To test the significance of these effects, we utilized 5000 bootstrapping samples with 95% bias-corrected confidence intervals. The bootstrapping is an efficient method to test significance in mediation models, since it makes no assumptions regarding the normality of distribution of the tested variables [26]. This assumption of normality is often violated while examining indirect effects (i.e., the product of the coefficients for paths *a* and *b*) in mediation models. Furthermore, *p*-values less than or equal to 0.05 were considered significant.

## 3. Results

### 3.1. Differences in Post-Traumatic Growth and Resilience According to General Characteristics

Table 1 shows respondents’ general characteristics. All respondents were in their early twenties (with a mean average of 22 years), consisting of 277 female students (95.2%) and 14 male students (4.8%). Most students were in their third year of study (68.4%; *n* = 199) and the rest were in their fourth year (31.6%; *n* = 92). The majority of the students (82.5%) had a mid-level economic status. For college major satisfaction, 56.4% of the nursing students considered themselves satisfied with their chosen major, and 42.6% were satisfied with university life. Further, 57.7% (*n* =168) students considered their clinical practice stress levels to be high.

The proportion of students with extrinsic motivation was higher (58.9%) than that of students with intrinsic motivation (41.1%). The most frequent extrinsic motivation was the ease of getting a job (40.5%), followed by nursing major choice based on high school grades alone (15.5%). The most frequent intrinsic motivation was students’ consideration of their aptitude and interest (29.9%), followed by their desire to serve and help other people (7.9%).

The PTG scores were significantly higher for students with intrinsic motivation than those with extrinsic motivation (*t* = 4.62, *p* < 0.001). Resilience variable showed the same tendency as PTG and the resilience scores showed similar results (*t* = 3.81, *p* < 0.001). The PTG scores showed significant differences according to: stress of clinical practice (F = 10.72, *p* < 0.001; nursing major satisfaction (F = 41.37, *p* < 0.001); and satisfaction with university life (F = 24.82, *p* < 0.001). Furthermore, post hoc tests showed that the lower the nursing students’ stress level and the higher their satisfaction with the chosen major and university life, then the higher the PTG. 

### 3.2. Correlation Analysis between Main Variables

The analysis revealed that a correlational relationship among academic motivation, resilience, and PTG was statistically significant (Table 2). Specifically, the three variables of intrinsic motivation, resilience, and PTG showed positive correlations. In addition, when the intrinsic motivation was stronger in comparison to the extrinsic motivation, the scores of PTG and resilience were shown to be higher.

### 3.3. Mediating Effects of Resilience on Academic Motivation and Post-Traumatic Growth 

The results showed resilience mediated the relationship between academic motivation and PTG. A three-step mediation analysis, as described by Baron and Kenny, was used to test the mediating effects of resilience on the relationship between academic motivation and PTG (Table 3 and Figure 1). In Step 1, academic motivation, an independent variable, had a significant effect on PTG, a dependent variable (B = 0.27, *p* = 0.004). In Step 2, academic motivation had a significant effect on resilience, a mediator variable (B = 0.14, *p* = 0.007). In *Step 3*, resilience had a significant impact on PTG (B = 0.82, *p* = 0.000), and its mediating effect was verified. Additionally in Step 3, academic motivation had a marginally significant effect on PTG (B = 0.16, *p* = 0.060). Furthermore, with a decrease in the non-standardization coefficient (from 0.27 to 0.16), academic motivation was shown to have a mediating effect on PTG.

A significant total, direct, and indirect effects were also found between academic motivation and PTG (Table 4). Resilience mediated 40.9% of the relationship between the other two variables.

## 4. Discussion

There are individual differences among nursing students based on their own characteristics as well as various personal factors and attributes which help them manage adverse situations and achieve personal growth during clinical practice [7,21]. This study was performed to explore the relationships among PTG, academic motivation and resilience in nursing students, and identify factors that may influence PTG.

This study demonstrated that nursing students with higher intrinsic motivation—including aptitude, interest, and the desire to help others—reported significantly higher PTG and resilience than those with higher extrinsic motivation, such as the ease of finding employment and college major selection based on high-school grades. A correlation analysis revealed that the stronger the students’ intrinsic motivation, the higher the PTG and resilience scores were for those students. Furthermore, nursing students’ intrinsic academic motivation seemed to influence their PTG and resilience; and resilience mediated 40.9% of the relationship between academic motivation and PTG. 

The results of a few previous studies have shown that intrinsically motivated nursing students were more likely to be satisfied with their chosen major, university life, and self-directed learning ability than extrinsically motivated students [27], and intrinsic motivation was also shown to reduce test anxiety [28]. There is little research on the relationship between academic motivation and PTG among nursing students, therefore making any direct comparison difficult. Previous results seemed consistent with this study, in that students with high scores of intrinsic motivation showed higher scores of PTG and satisfaction with their majors and university life, and exhibited lower scores in clinical practice stress.

Since students who selected their majors based on intrinsic motivation were most likely to be interested in their majors, they tended to expend a significant effort and attention on learning. They also responded more flexibly to stressful situations, and improved their resilience, which enabled them to successfully overcome perceived difficulties during their clinical practice [29]. As a result, they would achieve higher PTG leading to positive changes in themselves, interpersonal relationships, and life philosophy. 

However, this study showed that 58.9% of the participants selected nursing as their major due to extrinsic, rather than intrinsic motivation. The most frequent extrinsic motivation was the perceived ease of getting a job (40.5%), followed by choosing nursing major based on high-school grades (15.5%). Previous studies have shown that more than half of nursing students reportedly selected their major due to external motivation, rather than intrinsic motivation [27], which is in line with the findings of the current study. Another previous study reported that in South Korea the primary reason nursing students chose their major was employability (55%), followed by aptitude (20%); whereas, non-nursing students cited aptitude (48%), and employability (23%) as the most important reasons for their major choice [21]. Comparatively, an Australian study [30] reported that most students chose nursing as their major due to intrinsic motivation, such as their desire to care for people or enjoyment of nursing, rather than a career-related motivation. 

These results seem to be related to poor retention rates of nurses in South Korea. The resignation rate for newly graduated nurses within one year was reported to reach 42.7% [2], and it is estimated that nearly half of the nurses resigned due to maladjustments [31]. To address this problem, the South Korean government is attempting to increase the number of student admissions to nursing schools [32]; however, the shortage of nursing personnel is still a serious problem in South Korea [33]. 

Many South Korean college students are choosing nursing mainly due to the ease of obtaining employment related to the current economic recession and job shortages [21]. However, many nursing students who were extrinsically motivated in their major choices tended to drop out from the nursing program, which was associated with maladjustment after being admitted to nursing school or the inability to overcome work-related stress once employed [27]. The findings of this study supported that PTG levels were lower for students with higher extrinsic motivation (compared to those with higher intrinsic motivation) along with higher perceived stress in clinical practice, and lower satisfaction with their chosen major and college life. This phenomenon not only threatens the mental health and well-being of nursing students (and employed nurses upon graduation), but also can possibly impact the high resignation rate and quality of nursing care. 

In addition, many Korean parents influence the decision-making process of their children during the selection of a university and major field of study. They tend to persuade their children to choose a university and major that guarantee financial independence and social success. For this reason, there are many university students in Korea who choose their majors without considering their aptitudes. Therefore, many who choose nursing because of parents’ recommendations or guaranteed employment face ambivalence in regard to their career, which can lead to dropping out of the program or resigning from their nursing jobs [34].

It is imperative for nurse educators to be aware of nursing students’ motivation on choosing their majors. Based on this study and other similar research results, nursing students with low intrinsic motivation should be considered a high-risk group and strategic programs should be designed to increase retention rates. Nurse educators should also provide various opportunities for nursing students to better adjust and positively experience the core values of nursing in their school programs [35]. 

Although intrinsic motivation was essential to achieve PTG, resilience mediated 40.9% of the relationship between academic motivation and PTG, which indicated that resilience is also an important variable for PTG. Several earlier studies also reported that resilience positively affects PTG [36,37,38]. Resilience is an important attribute for nursing students and also nurses, since they often face adversity in clinical practice settings. Even though resilience is considered a personal trait, research provides strong evidence that resilience can be acquired through learning [39]. Effective mentoring, reflection workshops, reflective meditation, and a positive mindset toward university life can be used to increase nursing students’ resilience [40,41,42]. 

Additionally, cognitive processing is important for improving intrinsic motivation and resilience to achieve PTG. Earlier studies found that not all highly resilient people acquire high PTG, and that cognitive processing is necessary for personal growth [20,37,38]. Tedeschi and Calhoun [43] suggested that cognitive processing of a traumatic event, particularly a process of ruminative thought, is associated with this growth. They also suggested that although people experience automatic and intrusive rumination related to negative emotions soon after a traumatic event, these emotions can change to be a more deliberate rumination; this facilitates a positive schema change, such as finding strategies to reduce emotional distress. 

Many earlier studies have reported positive relationships between deliberate rumination and PTG [44,45]. In particular, several studies have examined the effects of programs that enhance deliberate rumination and improve PTG such as mind subtraction meditation [46,47], mindfulness-based intervention [35], and group reminiscence programs [48]. People attending a program that used rumination as one of the strategies were reported to start perceiving themselves differently, finding new meaning in their lives, applying new priorities, and increasing their appreciation of life [49].

This cognitive processing is aided by self-disclosure in supportive social environments [37,38,43]. Moreover, students’ resilience and internal growth can improve in this supportive atmosphere, where they share their experiences and discover positive meanings [50,51]. Therefore, in order to enable nursing students to adapt successfully and achieve PTG, supportive self-reflective programs promoting rumination and resilience in a positive atmosphere should be included in the nursing curriculum. 

Compared with results in Hwang’s study [7], our results showed there was no gender difference in the level of resilience and post-traumatic growth. However, it is difficult to generalize these findings according to gender considering the relatively low proportion of male students (just 4.8% of the total sample in this study). The number of male nursing students is continuing to increase, so it might be necessary to conduct further studies with a similar number of male and female students to clarify the gender issue. 

This study had a few limitations. First, it used a convenience sample that may not be representative of nursing students of South Korea; and, therefore, the results are not generalizable. Second, this cross-sectional study used self-report questionnaires to collect data, which can be associated with a potential risk of response distortion. Third, although we suggested that the cognitive process of deliberate rumination could be important for achieving PTG, this was not examined in this study. Fourth, intrinsic motivation and extrinsic motivation were classified by using a questionnaire composed of only five questions to investigate the academic motivation of nursing students.

Despite these limitations, this study also has several significant implications. First, this study demonstrated that nursing students’ clinical practice experiences were not merely stressful, but could be traumatic enough to facilitate PTG, similar to that experienced by other healthcare professionals. Another notable implication is that this study uniquely investigated the relationships among academic motivation, resilience, and PTG among nursing students. It adds to the current scant literature on academic motivation, PTG, and resilience in nursing students by showing that PTG can be achieved when intrinsic motivation is high, and that resilience mediated 40.9% of the relationship between academic motivation and PTG. Finally, this study confirmed that more than half of South Korean nursing students had extrinsic motivations, rather than intrinsic motivations, for choosing nursing as their major. We also found that low PTG and resilience were associated with high clinical practice stress, and low satisfaction with one’s major and university life. 

Based on our findings, we suggest a program enhancing nursing students’ ability to achieve PTG by reinforcing their intrinsic motivation and resilience should be offered as part of the curriculum. One-on-one or group mentoring, reflection workshops promoting rumination, and self-reflective meditation could be some of the approaches used when developing it. In addition, nursing students’ motivation on choosing their major need to be examined during their early college years. Through this strategy, they could overcome the stress and challenges of their clinical practice, increase their appreciation for life, and match their own values to their chosen major and job identity. Effective strategies are also needed to optimally self-select a career path; and proper training and education should be provided to ensure fulfilment of students’ career preference and motivation [12,27,51]. 

Finally, we suggest some directions for future research. First, nursing students’ characteristics need to be explored in a descriptive study with much larger samples in order to better understand the unique characteristics of South Korean nursing students and explore more possible predictors of PTG. Secondly, programs to enhance PTG, and to improve intrinsic motivation and resilience in nursing students need to be developed; and these programs would have to be examined for their effectiveness. Thirdly, we suggest that more studies are needed to explore the relationship between nurse retention rates and academic motivation. 

## 5. Conclusions

The findings of the current study suggested that in South Korea, more than a half of nursing students chose their major based on extrinsic rather than intrinsic motivation. The nursing students with high intrinsic academic motivation were more likely to achieve PTG, and resilience mediated the relationship between academic motivation and PTG. In light of this finding, the importance of academic motivation must be considered in the early stages of university-level nursing education. Nursing education strategies which help to strengthen students’ intrinsic motivation should be offered. Additionally, in order to improve resilience and PTG in nursing students, effective programs should be designed and provided with appropriate use of mentoring, workshops, self-reflection, and deliberate rumination.

## Figures and Tables

**Figure 1 ijerph-17-04901-f001:**
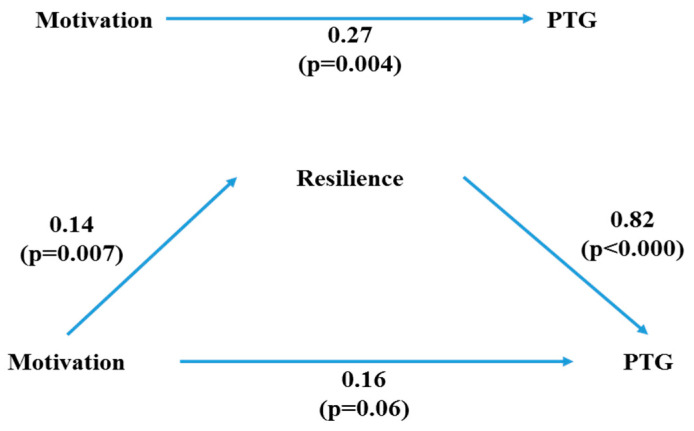
Resilience mediation models of the relationship between academic motivation and post-traumatic growth. All Steps included age, gender, school year, self- reporting economic status, grade point average, stress level of clinical practice, satisfaction with ones’ major and satisfaction with university life as covariates.

**Table 1 ijerph-17-04901-t001:** Differences in post-traumatic growth (PTG) and resilience according to general characteristics.

Variables	Category		*n* (%) or Mean (SD)	PTG (0–5)		Resilience (1–5)	
				Mean (SD)	t/F(*p*)	Mean (SD)	t/F(*p*)
Age	Range (20–27)		22.1(1.36)				
Gender	Female		277 (95.2)	2.93 (0.84)	1.08	3.61 (0.45)	0.51
	Male		14 (4.8)	3.18 (0.74)	(0.280)	3.67 (0.44)	(0.607)
School year	Third		199 (68.4)	3.00 (0.79)	1.80	3.60 (0.46)	−1.17
	Fourth		92 (31.6)	2.81 (0.94)	(0.071)	3.66 (0.43)	(0.241)
Self-reporting economic status	High ^a^		31 (10.7)	3.04 (0.94)	0.47 (0.622)	3.78 (0.46)	3.27 *
	Middle ^b^		240 (82.5)	2.94 (0.83)		3.60 (0.44)	(0.039)
	Low ^c^		20 (6.8)	2.81 (0.87)		3.48 (0.39)	a, b > c
Grade point average	4.0 or More		177 (60.8)	3.00 (0.86)	1.41 (0.246)	3.67 (0.43)	2.92 (0.055)
	3.0–3.9		102 (35.1)	2.83 (0.79)		3.54 (0.41)	
	2.9 or Less		12 (4.1)	3.07 (0.84)		3.53 (0.72)	
Academic motivation	Intrinsic	Consider aptitude and interest	87 (29.9)	3.15 (0.73)	4.62 (0.000)	3.72 (0.39)	3.81 (0.000)
		Can serve and help other people	23 (7.9)	3.43 (0.66)		3.75 (0.41)	
		Capable of working in foreign countries or studying continuously	9 (3.1)	3.22 (1.11)		3.77 (0.57)	
		Total	119 (41.1)	3.21 (0.75)		3.73 (0.41)	
	Extrinsic	Ease of getting a job after graduation	118 (40.5)	2.78 (0.77)		3.48 (0.40)	
		Choice based on high school grades only	45 (15.5)	2.66 (0.92)		3.57 (0.51)	
		Other	9 (3.1)	3.30 (0.77)		3.98 (0.59)	
		Total	172 (58.9)	2.76 (0.85)		3.53 (0.44)	
Stress level of clinical practice	High ^a^		168 (57.7)	2.79 (0.86)	10.72 *	3.58 (0.44)	6.70 *
	Middle ^b^		98 (33.7)	3.06 (0.75)	(0.000)	3.60 (0.43)	(0.001)
	Low ^c^		25 (8.6)	3.54 (0.74)	c > a, b	3.92 (0.38)	c > a, b
Satisfaction with one’s major	Satisfied ^a^		164 (56.4)	3.21 (0.71)	41.37 *	3.76 (0.39)	24.13 *
	Fair ^b^		84 (28.9)	2.89 (0.69)	(0.000)	3.49 (0.39)	(0.000)
	Unsatisfied ^c^		43 (14.8)	2.05 (0.94)	a > b>c	3.32 (0.50)	a > b>c
Satisfaction with university life	Satisfied ^a^		124 (42.6)	3.26 (0.72)	24.82 *	3.78 (0.42)	17.55 *
	Fair ^b^		105 (36.1)	2.90 (0.74)	(0.000)	3.54 (0.41)	(0.000)
	Unsatisfied ^c^		62 (21.3)	2.40 (0.95)	a > b > c	3.41 (0.43)	a > b, c

Note. SD, standard deviation. * Scheffè post hoc test (^a,^
^b,^
^c^ means each group compared in Scheffè test).

**Table 2 ijerph-17-04901-t002:** Correlations among motivation, post-traumatic growth, and resilience (*n* = 291).

Variables	Mean ± SD	Academic Motivation r (*p*)	Post-Traumatic Growth r (*p*)	Resilience r (*p*)
Academic motivation	0.41 ± 0.49	1		
Posttraumatic growth	2.94 ± 0.84	0.210 (<0.001)	1	
Resilience	3.61 ± 0.44	0.173 (<0.001)	0.361 (<0.001)	1

Note. SD, standard deviation. Academic motivation (0, extrinsic; 1, intrinsic).

**Table 3 ijerph-17-04901-t003:** Mediating effect of resilience.

Variable	Step 1. PTG					Step 2. Resilience					Step 3. PTG				
	B	S.E.	β	t	*p*	B	S.E.	β	t	*p*	B	S.E.	β	t	*p*
Constant	−0.86	0.28		−3.04	0.003	−0.49	0.15		−3.24	0.001	−0.45	0.26		−1.75	0.081
Age	0.05	0.04	0.09	1.35	0.180	0.00	0.02	0.01	0.21	0.833	0.05	0.04	0.08	1.39	0.165
Gender	−0.06	0.23	−0.02	−0.28	0.779	−0.04	0.12	−0.02	−0.36	0.722	−0.03	0.21	−0.01	−0.14	0.890
School year	−0.15	0.11	−0.08	−1.33	0.184	0.13	0.06	0.14	2.19	0.030	−0.26	0.10	−0.14	−2.54	0.012
Self-reporting economic status	−0.01	0.18	0.00	−0.06	0.952	0.11	0.10	0.06	1.15	0.253	−0.10	0.16	−0.03	−0.63	0.530
Grade point average	0.06	0.09	0.03	0.59	0.557	0.04	0.05	0.05	0.85	0.396	0.02	0.09	0.01	0.24	0.814
Stress level of clinical practice	0.22	0.05	0.22	3.96	0.000	0.05	0.03	0.10	1.85	0.065	0.17	0.05	0.17	3.48	0.001
Satisfaction with one’s major	0.34	0.11	0.20	3.17	0.002	0.20	0.06	0.22	3.42	0.001	0.18	0.10	0.10	1.80	0.073
Satisfaction with university life	0.27	0.10	0.16	2.63	0.009	0.15	0.05	0.17	2.84	0.005	0.14	0.09	0.08	1.51	0.133
Academic motivation	0.27	0.09	0.16	2.91	0.004	0.14	0.05	0.15	2.69	0.007	0.16	0.09	0.09	1.89	0.060
Resilience											0.82	0.10	0.43	8.17	0.000

Note. The reference groups for the table data are as follows: gender (male), academic year (third), economic status (low group), grade point average (lower than 3.5), clinical stress level (low or middle), satisfaction with one’s major (unsatisfied or fair), satisfaction with university life (unsatisfied or fair), and academic motivation (extrinsic). PTG: post-traumatic growth., S.E.: standard error, β: standardized beta coefficient, B: non-standardized coefficients.

**Table 4 ijerph-17-04901-t004:** Total, direct, and indirect effects of motivation on post-traumatic growth (mediation model).

Direct Effect (CI)	Indirect Effect (CI)	Total Effect (CI)	Percentage of Total Effect Mediated (%)
0.162 (−0.017, 0.319)	0.112 (0.034, 0.204)	0.274 (0.084, 0.049)	40.9

Note. non-standardized B coefficients. 1. Data have been adjusted for age, gender, school year, economic status, grade, stress, study satisfaction, and life satisfaction. 2. Bootstrap confidence intervals (CIs) were constructed using 5000 resamples and 95% bias-corrected CIs.

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
