# Peer review of "Effects of Academic Motivation on Clinical Practice-Related Post-Traumatic Growth among Nursing Students in South Korea: Mediating Effect of Resilience"

_ijerph, 2020, doi:10.3390/ijerph17134901_

Round 1
Reviewer 1 Report
Dear authors
- Please describe the randomization procedures.
- Please think about the revision of the manuscript title and discussion, while most of the participants were females. This fact could significantly contribute to their behavior and motivation, and thus needs to be mentioned and discussed.
Reviewer 2 Report
Thank you for the opportunity to review your paper. I find it very well-written. It flowed nicely, and oftentimes questions I would think of would be answered by you all. I appreciate the acknowledgment of limitations of your study as well as suggestions for future directions in research as well as education. I would be interested in seeing a longitudinal study looking at extrinsic versus intrinsic motivation for entering nursing. For example, do students respond the same when in their 1st year as they now are in their 3rd and 4th years?
Reviewer 3 Report
First of all I would like to convey my congratulations to the research team for the initiative in this study.
Next, a comment and a suggestion:
- On line 30 "post-traumatic growth" does not appear in the descriptors as a MeSH term.
-As a personal suggestion, Tables 2 and 3 can benefit from the differences by category shown in Table 1. In particular for the variable Academic Motivation, which is subdivided into the intrinsic-extrinsic categories, as it corresponds to what is described, analyzed and discussed in the text.
Best regards
Reviewer 4 Report
This is an interesting paper about an important subject. I have just a few issues:
1. Although we understand that the authors are very interested in Korea, it would be mocre interesting for non-Korean readers if the authors could include some information about the situation (concerning drop-out rates, for example) among the nursing profession elsewhere. That way, the authors' very interesting and relevant results would seem more useful to non-Korean readers.
2. It would be useful to have slightly more information about the tools used (e.g. the questionnaires).
3. The authors could consider presenting some of their results visually, rather than in these dry tables.
4. There are a few mistakes in the English (e.g. "half of nursing students"). The authors could get the text checked again.
Reviewer 5 Report
General comments:
I believe that the problem of the article is relevant and current in nursing students.
- Title
The title reflects the content and problem studied.
- Abstract
The abstract meet the standards of the Journal. reference is made to the sample, methodology used (namely the research carried out, the inclusion criteria, the research platforms, etc.), the main results and conclusions. The implications of the investigation are addressed.
- Key Words
The keywords are representative of the subject studied and exposed, but resilence appears on the MeSH platform as Psychological Resilience
- Background
A state of the art is made in relation to the study. The objective of the study is mentioned, as well as the justification for the choice and importance of studying this theme
- Methods
There is detailed description of the research methods used. The design is correct and it is possible to validate the veracity of the results
The sample is insufficient as they have explained in the limitations
I don´t feel qualified to judge about the statistical analysis because of its complexity and there are the multiple analyses made.
I suggest a review by an expert
- Findings
The results shown are concrete and detailed, explaining how to obtain this information and what scientific evidence it has.
- Discussion
the discussion is extensive and reasoned
- Application to Pratice
The practical application of this investigation is explained.
- References
The references used are current, the vast majority dating from the last ten to twelve years.
